# In Vitro Investigation of the Impact of Bacterial–Fungal Interaction on Carbapenem-Resistant *Klebsiella pneumoniae*

**DOI:** 10.3390/molecules27082541

**Published:** 2022-04-14

**Authors:** Hani Moubasher, Amani Elkholy, May Sherif, Mariam Zahran, Sherif Elnagdy

**Affiliations:** 1Department of Botany and Microbiology, Faculty of Science, Cairo University, Giza 12613, Egypt; mariam.a.zahran@gmail.com (M.Z.); sh.elnagdy@gmail.com (S.E.); 2Department of Clinical Pathology, Faculty of Medicine, Cairo University, Giza 12613, Egypt; aakholy@gmail.com (A.E.); dr.maysherif@gmail.com (M.S.)

**Keywords:** virulence factors, co-culture, mutation, gene deletion, bacterial–fungal interaction, carbapenem resistance

## Abstract

Fungal–bacterial co-culturing is a potential technique for the production of secondary metabolites with antibacterial activity. Twenty-nine fungal species were screened in a co-culture with carbapenem-resistant *Klebsiella pneumoniae* at different temperatures. A temperature of 37 ° showed inhibition of bacterial growth. Antimicrobial susceptibility testing for *K. pneumoniae* was conducted to compare antibiotic resistance patterns before and after the co-culture. Genotypic comparison of the *K. pneumonia* was performed using next generation sequencing (NGS). It was shown that two out of five *K. pneumoniae*, with sequence type ST 101 isolates, lost *bla-_OXA48_*, *bla-_CTX-M-14_*, *tir*, *strA* and *strB* genes after the co-culture with *Scopulariopsis brevicaulis* fungus. The other three isolates (ST 383 and 147) were inhibited in the co-culture but did not show any changes in resistance. The total ethyl acetate extract of the fungal–bacterial co-culture was tested against *K. pneumoniae* using a disc diffusion method. The concentration of the crude extract was 0.97 mg/µL which resulted in total inhibition of the bacteria. Using chromatographic techniques, the purified compounds were identified as 11-octadecenoic acid, 2,4-Di-tert-butylphenol, 2,3-Butanediol and 9-octadecenamide. These were tested against *K. pneumoniae* using the well diffusion method at a concentration of 85 µg/µL which resulted in total inhibition of bacteria. The co-culture results indicated that bacteria under chemical stress showed variable responses and induced fungal secondary metabolites with antibacterial activities.

## 1. Introduction

Resistance to multiple classes of antibiotics—especially resistance to carbapenems—is one of the urgent threats identified by the World Health Organization (WHO) because carbapenems are considered the drug of last resort against infections caused by the Gram-negative pathogen (GNP) [1]. One of the most important pathogens that has shown notably high patterns of resistance in healthcare settings is Klebsiella pneumoniae (*K. pneumoniae*) [2].

Carbapenem-resistant *K. pneumoniae* is a GNP responsible for fatal infections acquired in hospitals in critically ill patients. These are patients who require prolonged hospital stays and repeated antibiotic therapy resulting in the emergence of flora resistance shifts, from normal sensitive flora to resistant flora [3]. Antimicrobial resistance (AMR) rates to beta-lactam antibiotics (penicillins, cephalosporins, monobactams and carbapenems) have increased particularly among healthcare-associated infections by GNP in developing countries. This is due to the lack of national antimicrobial stewardship programs, the widespread abuse of antimicrobials, including carbapenems in hospitals, and poor compliance with infection control practices. This has been a serious challenge in Egyptian hospitals [4,5].

The genes responsible for carbapenem resistance that were identified in Egypt included *bla_OXA_*, *bla_NDM_*, *bla_VIM_*, *bla_IMP_*, *and bla_KPC_* [6]. Extended spectrum beta-lactamase (ESBL) genotypes included *bla_CTX-M15_*, *bla_TEM-OSB_*, *bla_SHV-OSBL_*, *and bla_CTX-M-14_* [7,8]. A recent multicenter study from Egypt on 39 multi-drug resistant (MDR) *K. pneumoniae* isolates using NGS revealed that the most common acquired resistance gene found was *bla_CTX-M15_*, detected in 69.2%, and carbapenemase genes were found in 74.4%. The most common carbapenemase genes were *bla_NDM_* (35.9%), *bla_OXA-48_* (35.9%), and *bla_KPC_* (2.6%). Seven strains (18%) contained more than a single carbapenemase gene. Yersiniabactin was the most common virulence factor (69.2%). Hyper-mucoviscosity was documented for 6 out of 39 isolates [9].

To overcome infections, the host body has defense factors against bacterial infections. These factors include physical barriers like skin and mucous membranes, along with chemical factors. Once the bacteria invade the host cell, they develop virulence mechanisms to sustain their growth in adverse conditions [10]. Acquiring iron from the host environment is one of the most important virulence factors [9]. A range of *Enterobacteriaceae* encompass a gene cluster called the high-pathogenicity island that codes for proteins of the yersiniabactin siderophore and its uptake system [11], which enables the bacterial cell to acquire iron from host cells through a set of genes. These include *ybtA*, *ybtE*, *ybtP*, *ybtQ*, *ybtS*, *ybtT*, *ybtU*, and *ybtX* [12].

Another crucial virulence factor is the biofilm formation that is linked to chronic bacterial infections. Infections caused by *K. pneumoniae* strains that are able to form biofilms are more challenging to treat. Diago-Navarro E. and his colleagues [13] found that approximately half of the 40 studied carbapenem-resistant *K. pneumoniae* bacteremia strains were able to form biofilms. The genes that code for biofilm formation in *K. pneumoniae* include *magA*, *aero*, *rmpA*, *rmpA2*, *allS*, *wcaG*, *and iutA* [14]. As stated by Hancock V and his colleagues [15] *fyuA* gene encoding the yersiniabactin receptor, is one of the most upregulated genes in biofilm formation in urinary tract infections caused by *E. coli.*

In spite of the global antimicrobial resistance threat, the discovery of new antibiotics has not kept pace with the continuous spread of antibiotic-resistant infectious microorganisms, calling for an urgent search for natural compounds with antibacterial effect [16].

In nature, microorganisms exist in a community; for survival, one microbe could produce biological products to inhibit or kill other microbes in the battle for limited nutritional resources and the competition for space. Therefore, co-cultures of microorganisms in the same confined environment results in the production of potentially novel compounds through the stimulation of the silent genes of one partner or increasing the production of secondary metabolites [17]. Microorganism co-cultures can be achieved in either solid or liquid media and have recently been used extensively to study natural interactions and discover new bioactive metabolites [18]. Successful co-culturing experiments can be mediated through direct contact between the members of the co-culture, where the fungal mycelia and bacterial cells of both microorganisms interact together [19]. Interactions of microorganisms with the external environment is mediated through the cell wall, protecting the cells from oxidative or osmotic stresses, and modulating the responses to antimicrobial drugs [20]. Fungi are capable of adapting their cell walls in response to stress by activating several mechanisms directed towards healing or as compensation for cell wall damage [21].

Antimicrobial compounds produced by fungi are generally secondary metabolites that are not used for the growth of the fungi, but for self-defense and competition with other microbes to obtain nutrients, habitats, oxygen, light, and other growth requirements [22].

The most studied category of this interaction is antibiosis, which led to the development of several important antibiotics like the beta-lactam antibiotic, penicillin, that was developed as the result of antibiosis between *Staphylococcus* and *Penicillium* [23]. Several studies of fungal–bacterial interactions proved that co- culturing is a potential method for the induction of secondary metabolites [19], yet very little is known about the underlying molecular mechanisms of these interactions.

This study was conducted to utilize the potential of bacterial–fungal interaction in controlling carbapenem-resistant *K. pneumoniae* in vitro through the co-culture of fungi and *K. pneumonia* in both solid and liquid media.

## 2. Material and Methods

### 2.1. Isolation, Identification, and Antimicrobial Susceptibility Testing of K. pneumoniae

*K. pneumoniae* were isolated from blood cultures of ICU patients who had been diagnosed with bacteremia and sepsis in a tertiary care hospital in Egypt. Samples of 10 mL of peripheral blood were withdrawn aseptically from ICU patients with elevated sepsis markers and inoculated into BACT/ALERT^®^ blood culture bottles (Biomerieux, France). Positive blood culture bottles were sub-cultured on blood agar and MacConkey agar plates to isolate the pathogens. Identification of the isolated bacteria was performed by VITEK-2^®^ (Biomerieux, France) according to the published guidelines of the Clinical Laboratory Standards Institute (CLSI, 2019).

Antibiotic susceptibility testing was performed by disc diffusion method according to guidelines of the (CLSI, 2019).

### 2.2. Isolation, Screening, and Indentification of Fungi with Activity against K. pneumoniae

Fungi were isolated from the soil and hospital inanimate environment and the fungal isolates were screened for antibacterial properties against *K. pneumoniae*. Soil samples were collected from the garden of Cairo University and the hospital garden, at a depth of 10 cm below soil surface. An amount of 5 g of soil was added to 1 L of sterile water and a serial dilution was made [24]. From each dilution, 1 mL of spore suspension was cultured on Czapek’s Dox sterile agar media. The plates were incubated at 26 °C for 5 days [25]. Environmental swabs were collected from inanimate objects from empty rooms by wetting sterile cotton swabs with sterile saline and rubbing a 10 cm² surface of the tested area. The swabs were streaked on Czapek’s Dox agar and incubated at 26 °C for 5 days. Fungal isolates were identified in Assiut University Mycological Centre (AUMC, Assiut, Egypt).

### 2.3. Fungal Bacterial Co-Culture

Screening of 29 Fungal Isolates for Antibacterial Activity [26]. Fungal–bacterial co-culture was performed by preparing 0.5 McFarland standard (MCF) suspension from each *K. pneumoniae* isolate, streaking the suspension on Mueller–Hinton Agar plate (HIMEDIA Laboratories, India) using a sterile cotton swab and leaving the plate to dry for 5 min. A 0.5 cm in diameter fungal disc was collected from a 5-day-old fungal plate and placed in the center of the petri dish using a sterile needle. Two petri dishes 3X replicates were prepared for each *K. pneumoniae* isolate, with each of 29 fungal isolates, for incubation at two different temperatures (37 °C which is optimum for bacterial growth and 26 °C which is optimum for fungal growth). The plates were then examined daily for both fungal and bacterial growth

### 2.4. Subculture of K. Pneumoniae after Co-Culturing with Fungi

After 14 days of co-culture incubation, *K. pneumoniae* were sub-cultured from the zone around the fungal growth using a sterile loop and streaked on MacConkey agar (HIMEDIA Laboratories, India) and incubated for 24 h at 37 °C. Antimicrobial susceptibility testing was then performed by disc diffusion method to compare between the antimicrobial susceptibility of *K. pneumoniae* before and after the co-culture. The inhibited isolates were sub-cultured repeatedly on Mueller–Hinton agar. When retested for the susceptibility to carbapenems, they were found to remain carbapenem-susceptible.

### 2.5. Next Generation Sequencing

Next generation sequencing was used for whole-genome sequencing of *K. pneumoniae* isolates before and after the co-culture. The bacterial DNA was extracted from freshly sub-cultured colonies using the QIAamp DNA Mini Kit (Qiagen, cat # 51304) according to the manufacturer’s instructions, and the DNA concentration was measured using a Denovix Fluorometer (Denovix, Wilmington, DC, USA). The genomic DNA was stored at −20 °C. Total bacterial DNA (1 ng) was used in the library preparation. The library was prepared with Nextera XT DNA Library Preparation Kit (FC-131-1096, Illumina, San Diego, CA, USA), according to the manufacturer’s instructions. To summarize, transposons were used to fragment the DNA, subsequently, adapter sequences were added onto the DNA template. The product was then size-selected for optimum insert length, enriched, and quantified. Sequencing was carried out with the MiSeq reagent kit 600 v3 (Illumina, USA) on the Illumina MiSeq, generating, on average, 301 base pair paired-end reads.

FastaQ files were uploaded on Illumine BaseSpace and were assembled de novo using the Spades application. The assembled FASTA contig files were uploaded to both the PATRIC online tool and the Center of genomic epidemiology online website database for further analysis.

### 2.6. Preparation of Crude Extract of Fungal–Bacterial Co-Culture

The broth media was filtered using filter paper Whatmann No.1 to remove fungal mycelia and then centrifuged at 8000× *g* for 10 min to remove the bacteria. The supernatant was transferred to a separatory funnel mixed with same volume of dichloromethane 3 times. The funnel was strongly shaken and then left to allow partitioning. Ethyl acetate was added and the same process mentioned with dichloromethane was repeated. dichloromethane and ethyl acetate extracts were evaporated to dryness under reduced pressure using a rotary evaporator at temperatures of 45 °C and 60 °C, respectively, and then the 2 extracts were weighed separately.

### 2.7. Testing the Crude Extract against K. pneumoniae

After evaporation of both extracts, they were tested against *K. pneumoniae* isolates before co-culture using the disc diffusion method by adding 50 µL of each extract on 1 cm in diameter filter paper disc, which was then allowed to dry to assess their antibacterial effects at the same time MIC was performed.

### 2.8. Column Chromatography for Compound Elucidation

Sephadex^®^ LH-20 (Merk, Darmstadt, Germany) was used as stationary phase. The Sephadex was suspended in chloroform to pack the column. The solvent-resistant column was 40 cm long, with a diameter of 2.5 cm and had a glass stopper at the bottom. The Sephadex suspension was introduced gradually, and the final column size was 30 × 2.5 cm. A 243.3 mg ethyl acetate extract was dissolved in 5 mL of chloroform and was passed through the column with flow rate 0.2 mL/min. A gradient solvent system of cyclohexane–chloroform–methanol (3:1:1, 2:1:1, 1:1:1) was used. Fractions of 5 mL were collected.

Each fraction was dissolved in 200 µL ethanol and 10 µL were added to each well and tested against *K. pneumoniae* using the agar well diffusion method to detect which fraction expressed the most antibacterial activity

### 2.9. Gas Chromatography Mass Spectroscopy (GC-MS)

GC-MS analyses were performed with an injection volume of 2 μL and a split ratio of 30. Specifications for the measurements in DCM and ACE are as follows:Injector temperature 200 °C, Temperature program: T1 = 35 °C/3 min, R1 = 10 °C/min, T2 = 220 °CInjector temperature 280 °C, Temperature program: T1 = 35 °C/3 min, R1 = 10 °C/min, T2 = 300 °C

Specifications for the measurements in MET. 

### 2.10. Ethical Consideration

The study protocol was approved by the Research Ethics Committee of Cairo University Medical School in accordance with the Declaration of Helsinki (Ethical approval number: N-13-2020).

## 3. Results

Five carbapenem-resistant *K. pneumoniae* isolates were selected. They were also resistant to other antibiotics commonly used for treatment of patients with Gram-negative infections. The 5 isolates were referred to as K5, K92, K14, K15, and K16, and their sequence types were ST101, ST101, ST383, ST147, and ST383, respectively.

A total of 29 fungal isolates were isolated from the soil and from environmental swabs of the hospital, belonging to 11 fungal species. Table 1.

Fungal isolates that exhibited better growth in co-culture with *K. pneumoniae* were identified in Assiut University Mycological Centre (AUMC, Egypt). The fungal isolate that exhibited the best growth rate in co-culture with *K. pneumoniae* was *Scopulariopsis brevicaulis*, a common soil saprophyte and rare human pathogen [27]. Although the fungal growth at 26 °C was faster, it did not show a definite inhibitory effect on the bacterial growth. On the contrary, at 37 °C there was a decrease in the density of bacterial growth around the fungal inoculums (Figure 1). Accordingly, we completed the study at 37 °C.

*K. pneumoniae* that was isolated from the less dense zone around the fungal growth in the co-culture that after 14 days’ incubation at 37 °C as shown in Figure 2, and retested their antimicrobial susceptibility (AST) compared to the corresponding AST before co-culture. We noticed that both K5 and K92 became sensitive to Imipenem and Meropenem after the co-culture, while K14, 15 and 16 AST remained resistant to Imipenem and Meropenem after co-culture (Table 2).

Whole-genome sequencing was conducted for genotypic analysis of *K. pneumoniae* before and after co-culture (NCBI submission SUB9435847) and each *K. pneumonia* sample was given a genomic accession number as shown in Table 3 Thirty-one antimicrobial resistance genes (ARGs) were selected for comparison (Table 4). We compared those that confer resistance to beta-lactams (*bla**_CTX-M-14b_*, *bla_CTX-M-15_*, *bla_TEM-1B_*, *bla_OXA-48_*, *bla_SHV-1_*, *bla_SHV-12_*, *bla_NDM-5_* and *bla_OXA-9_*), those that confer resistance to aminoglycoside resistance (*aph**(3′)-Ia*, *aac(6′)Ib-cr*, *aph(3′)-Via*, *aph(3′)-VIb*, *aadA1*, *armA*, *strA*, and *strB*), those that confer resistance against quinolones (*oqxA*, *oqxB*, *QnrS1*, and *QnrB1*), and those that confer resistance to rifampicin, fosfomycin, macrolides, sulphonamides, trimethoprim, tetracycline and phenicol (*ARR-3*, *fosA*, *mph(A)*, *msr(E)*, *mph(E)*, *sul1*, *sul2*, *dfrA5*, *tet(A)* and *catA1*).

Comparing the antibiotic resistance, virulence factors, efflux pumps, and efflux pump-related products of five isolates before and after co-culture, we noticed that *bla_CTX-M-14b_*, *bla_OXA-48_*, *tir*, *strA*, and *strB* genes were lost from K5 and K92 after co-culture with the fungus. However, the resistance genes of K14, K15, and K16 showed no change after co-culture, nor did K5 show any change in virulence genes, efflux genes, or efflux- related products. K92 showed loss of the *pagO* gene, which belongs to virulence genes. It was observed that *fyuA* gene and the *ybtA*, *ybtE*, *ybtP*, *ybtQ*, *ybtS*, *ybtT*, *ybtU*, and *ybtX* set of genes belonging to virulence factors were lost from K16 after co-culture. We did not notice any gene loss in either K14 or K15.

Both ethyl acetate (EA) and dichloromethane (DCM) extracts (243.3 mg and 214.5 mg, respectively) were dissolved in 250 µL ethanol and 10 µL were added to each disc and then tested against *K. pneumoniae* using the disc diffusion method. The results showed a zone of inhibition around the EA disc while the DCM disc showed no zone of inhibition as shown in Figure 3 seven fractions were collected and then left to dry. Table 5 shows the dry weight in each fraction.

Fraction 2 showed significant inhibition in bacterial growth around the disc, as shown in Figure 4

Four different compounds were identified by GC-MS as 11-octadecenoic acid, 2,4-Di-tert-butylphenol, 2,3-Butanediol, and 9-octadecenamide, which showed complete inhibition of bacteria around the well.

## 4. Discussion

*K. pneumoniae* resistance has become a public health hazard worldwide. It has become gradually resistant to penicillin, aminoglycosides, extended-spectrum β-lactamase, and fluoroquinolones. This resistance is due to chromosomal mutations and the presence of many transmissible plasmids [28]. When carbapenem antibiotics were introduced in the 2000s as drugs of the last resort for the treatment of infection caused by extended-spectrum-lactamase (ESBL) producing Gram-negative bacteria, resistance to carbapenems emerged in strains that produced carbapenemases. High rates of *K. pneumoniae* were reported from Egypt [29], and in carbapenem-resistant *K. pneumoniae* isolates there was a high prevalence of *K. pneumoniae* carbapenemase (KPC), New-Delhi metallo beta-lactamases (NDM), and OXA-48 genes [30]. Facing such an urgent challenge, only a limited number of antimicrobials have been introduced into the market. Modern improvements in microbial genomics have proved that many microorganisms can produce natural products under specific laboratory conditions [31].

Previous studies showed remarkable fungal–bacterial interactions in co-cultures. Nogueira et al., 2019 [32], examined the interactions of *K. pneumoniae* and different *Aspergillus* species using co-cultures. The results showed that *K. pneumoniae* could inhibit spore germination, hyphal growth, and biofilm formation of *Aspergillus* species in vitro. The study also presented the importance of physical contact between fungi and bacteria for the effect to take place. A similar antagonistic effect of *K. pneumoniae* on biofilm formation of *Candida albicans* was reported by Fox et al., 2014 [33].

In our study, the antibacterial activity of fungal secondary metabolites was observed during the co-culture of *S. brevicaulis* with *K. pneumoniae* isolates. The results showed that the growth rate of the fungus was enhanced at 26 °C compared to its growth rate at 37 °C. This finding is in agreement with previous reports [34]. However, the co-culture at 37 °C showed a more clearly identifiable zone of lower density bacterial growth that was not observed at 26 °C.

It has been anticipated that bacterial population is made up of identical cells. However, according to Gómez, 2010 [35], genetically matching bacterial cells may have variable metabolism and growth rates along with other cell functions such as efflux pump and biofilm formation. Ackermann M, 2015 [36], referred to this phenomenon as ‘phenotypic heterogeneity’ and it has been detected in many different bacterial species.

A group of genes that belongs to the resistance nodulation superfamily of efflux pumps is considered a main reason for multi-drug resistance in bacteria [37]. By ejecting structurally unrelated antibacterial compounds, it decreases the intracellular concentration of the antibacterial agent [38]. Among those genes are AcrA and AcrB, which are responsible for a protein complex, TolC, that is located asymmetrically in the bacterial cell at the poles of rod-shaped bacteria such as those belonging to *Enterobacteriaceae* family [39]. The mother cell retains an old cell pole that originated in a past binary fission, which enables the mother cell to pump out drugs more efficiently than daughter cells, giving the mother cell the advantage of having a faster growth rate than daughter cells at low concentrations of antibacterial agents [40].

It was speculated that the *K. pneumoniae* that survived the presence of small concentrations of the antibacterial agent secreted from *S. brevicaulis* during co-culture had the same efflux pumping capacities as the mother cell, allowing their survival, while the daughter cells that lacked the same efflux pumping capacities ceased to survive causing the formation of the less dense zone of bacterial growth around the fungi at 37 °C.

The less dense zone of bacterial growth did not occur when the co-culture was allowed to grow at 26 °C. This is likely because the antibacterial agent secreted by *S. brevicaulis* was produced in much higher concentrations at 37 °C than at 26 °C. It is a well-established fact that fungi secrete secondary metabolites during stress conditions, and we assume that two stress factors were present in this study. The first stress factor being the high temperature at 37 °C, which is not favorable for fungal growth [41], and the second being the presence of bacteria in the same confined area [42]. To evaluate the effect of the co-culture on *K. pneumoniae* at the molecular level, a comparison of genes responsible for antibiotic resistance, virulence factors, efflux pumps, and efflux pump related products was conducted. The loss of carbapenem resistance was detected in two isolates (ST 101). In spite of the inhibited growth, the other three isolates (ST 383 and 147) remained carbapenem-resistant, suggesting different inhibition mechanisms.

The NGS analysis showed that after the co-culture with *S. brevicaulis*, the two *K. pneumoniae* strains, K5 and K92 (ST 101), lost the *IncL/M_(pOXA-48_)* plasmid, which harbors the gene responsible for resistance to carbapenem. This was confirmed by the AST of both strains before and after co-culture, as well as after sub-culturing a sample from the same plate after 14 days of co-culture away from the fungal growth. This result indicated the permanent loss of the plasmid.

The *bla_CTX-M-14b_* gene encoding for extended-spectrum beta-lactamase (ESBL) production may be subject to mutation or deletion after co-culture, although the bacteria remained an ESBL producer due to the presence of *bla_CTX-M-15b_* (which may be present on bacterial chromosome or on a different plasmid). There was also a loss of the *tir* gene that encodes a protein accountable for transfer inhibition through transposon Tn1999 and its alternatives [43].

Additionally, both the *strA* and *strB* genes, which are responsible for streptomycin resistance [44], were lost from both strains, while the other three did not show any change in resistance genes. Furthermore, the *pagO* gene that was lost from K92 is a putative membrane protein gene that belongs to the drug/metabolite transporter superfamilies [45]. Its main function is unknown.

Genomic analysis was performed for the other three *K. pneumoniae* strains (K14, K15 and K16) to explore virulence genes and efflux pump genes. K16, before and after the co-culture, revealed the loss of some of the following virulence genes that encode different components of the bacterial siderophore systems: *fyuA* gene, along with the *ybtA*, *ybtE*, *ybtP*, *ybtQ*, *ybtS*, *ybtT*, *ybtU*, and *ybtX* set of genes.

Iron is a crucial nutrient for most bacterial species as it is a cofactor for the electron transport [46]. These results show that K16 lost crucial vitality and virulence determinants after the co-culture. In concordance, Perry RD, 2011 [12], demonstrated that a number of virulence factors were important for the development of the bubonic plague caused by *Yersinia pestis* due to the siderophore-dependent iron transport system. Yersiniabactin encoded on a high pathogenicity island that is prevalent among many pathogenic bacteria of *Enterobacteriaceae* family. Previous research evaluated the dissemination of siderophores among *K. pneumoniae* clinical isolates and found that most *K. pneumoniae* isolates produce enterobactin, while a much smaller percentage produce either aerobactin or yersiniabactin [47,48]. Yersiniabactin is vital for the infectivity and pathogenicity of *K. pneumoniae* against a mammalian host, and the presence of yersiniabactin contributes to a more virulent phenotype of *K. pneumoniae* [49].

## 5. Conclusions

In this study, the molecular genotypes and phenotypes of antibiotic-resistant *K. pneumonia* were investigated before and after the co-culture with *S. brevicaulis*. The correlation between bacterial–fungal interaction, antibiotic resistance, and virulence factors in *K. pneumonia* were determined. Our results indicated that *K. pneumonia* ST101 lost *bla_CTX-M-15b_* and *bla_OXA-48_* genes and that the consortium of compounds were able to inhibit *K. pneumonia* isolates with ST101, 383, and 147, which could facilitate future attempts to control drug resistant bacteria.

Such findings may increase our understanding of the potential of bacterial–fungal interaction in the control of antibiotic resistance and in the production of compounds with antibacterial effects.

## Figures and Tables

**Figure 1 molecules-27-02541-f001:**
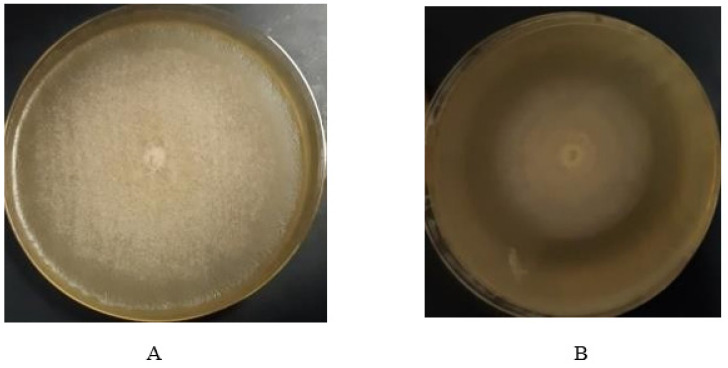
Co-culture of *Scopulariopsis brevicaulis* with *K. pneumoniae* at (**A**) 26 °C, and (**B**) co-culture with *K. pneumoniae* at 37 °C.

**Figure 2 molecules-27-02541-f002:**
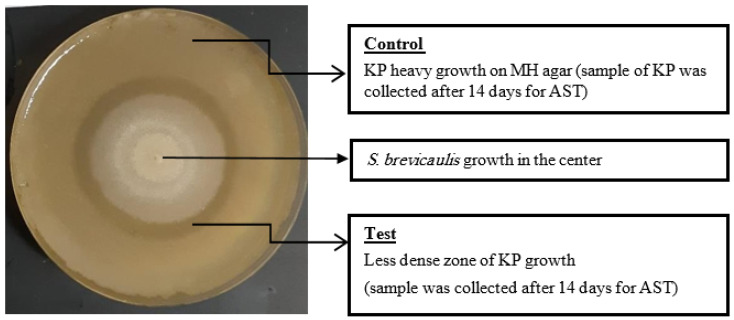
Co-culture of *Scopulariopsis brevicaulis* in co-culture with *K. pneumoniae* at 37 °C after 14 days of incubation.

**Figure 3 molecules-27-02541-f003:**
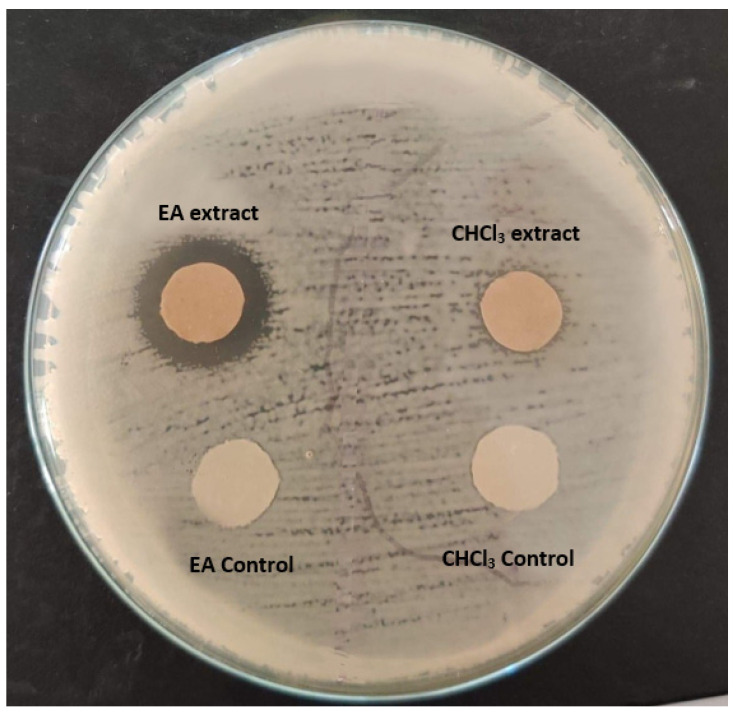
Testing total extract against *K. pneumoniae*.

**Figure 4 molecules-27-02541-f004:**
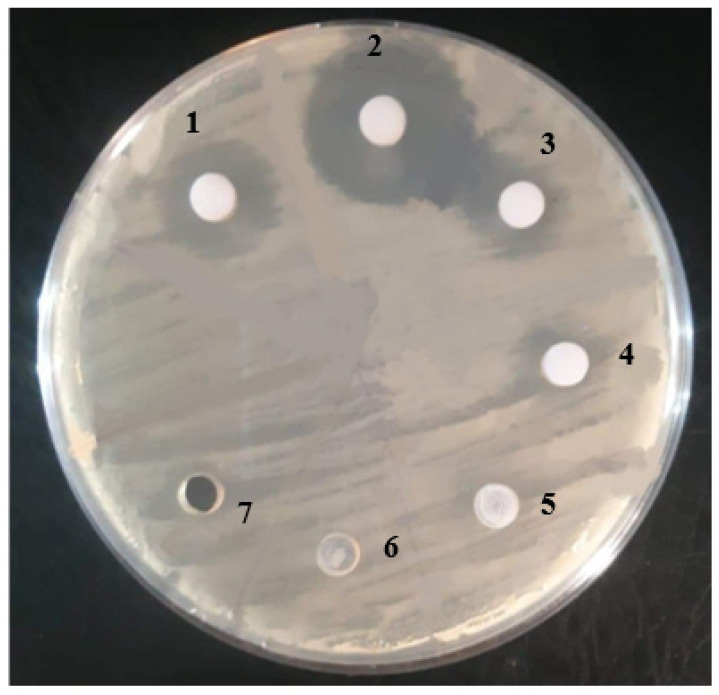
Testing fractions from column chromatography against *K. pneumoniae*.

**Table 1 molecules-27-02541-t001:** Identification of Fungal isolates.

AUMC No.	Identification
**1167**	*Scopulariopsis brevicaulis* **(Saccardo) Bainier**
**11678**	*Curvularia Lunata* **(Wakker) Boedijn**
**11679**	*Aspergillus niger* **van Tieghem**
**11554**	*Aspergillus flavus* **(Link)**
**11555**	*Curvularia brachyspora* **(Boedijn)**
**11556**	*Penicillium chrysogenum* **(Thom)**
**11557**	*Aspergillus flavus* **(Link)**
**11559**	*Penicillium chrysogepnum* **(Thom)**
**11561**	*Cladosporium sphaerospermum* **Penzig**
**11562**	*Alternaria alternate* **(Fries) Keissler**
**11564**	*Stemphylium botryosum* **Wallroth**

**Table 2 molecules-27-02541-t002:** Antimicrobial susceptibility test results using disc diffusion method of original *K. pneumoniae* isolates compared to the inhibited *K. pneumoniae* after the co-culture.

	K. pneumoniae	K5 before Co-Culture	K5 after Co-Culture	K92 before Co-Culture	K92 after Co-Culture	K15 before Co-Culture	K15 after Co-Culture	K14 before Co-Culture	K14 after Co-Culture	K16 before Co-Culture	K16 after Co-Culture
Antibiotic											
**Imipenem (IMP)**		11 mm (R)	25 mm (R)	9 mm (R)	23 mm (S)	9 mm (R)	9 mm (R)	10 mm (R)	10 mm (R)	8 mm (R)	8 mm (R)
**Meropenem (MEM)**		6 (R)	20 mm (R)	6 (R)	19 mm (S)	6 (R)	6 (R)	6 (R)	6 (R)	6 (R)	6 (R)
**Piperacillin/Tazobactam (TZP)**		6 (R)	20 mm (R)	6 (R)	21 mm	6 (R)	6 (R)	6 (R)	6 (R)	6 (R)	6 (R)
**Cefepime (FEP)**		6 (R)	12 mm (R)	6 (R)	12 mm	6 (R)	6 (R)	6 (R)	6 (R)	6 (R)	6 (R)
**Cefotaxime (CRO)**		6 (R)	6 (R)	6 (R)	6 (R)	6 (R)	6 (R)	6 (R)	6 (R)	6 (R)	6 (R)
**Cefotaxime (CTX)**		6 (R)	6 (R)	6 (R)	6 (R)	6 (R)	6 (R)	6 (R)	6 (R)	6 (R)	6 (R)
**Amoxicillin/Clavulanate (AMC)**		6 (R)	15 mm (S)	6 (R)	16 mm	6 (R)	6 (R)	6 (R)	6 (R)	6 (R)	6 (R)
**Ceftriaxone (CRO)**		6 (R)	6 (R)	6 (R)	6 (R)	6 (R)	6 (R)	6 (R)	6 (R)	6 (R)	6 (R)
**Trimethoprim-sulfamethoxazole (SXT)**		6 (R)	17 (R)	6 (R)	20 mm	6 (R)	6 (R)	6 (R)	6 (R)	6 (R)	6 (R)
**Ciprofloxacin (CIP)**		6 (R)	6 (R)	6 (R)	6 (R)	6 (R)	6 (R)	6 (R)	6 (R)	6 (R)	6 (R)

**Table 3 molecules-27-02541-t003:** Genome accession number for each *K. pneumoniae*.

Sample	Genome Accession
K5_BEFORE	JAHTMJ000000000
K5_AFTER	JAHTMI000000000
K14_BEFORE	JAHTMF000000000
K14_AFTER	JAHTME000000000
K15_BEFORE	JAHTMD000000000
K15_AFTER	JAHTMC000000000
K16_BEFORE	JAHTMB000000000
K16_AFTER	JAHTMA000000000
K92_BEFORE	JAHTMH000000000
K92_AFTER	JAHTMG000000000

**Table 4 molecules-27-02541-t004:** Comparison of the antimicrobial resistance genes before and after co-culture with five *K*. *pneumoniae*.

Resistance Genes	Phenotype	K5 before (ST 101)	K5 after (ST 101)	K92 before (ST 101)	K92 after (ST 101)	K14 before (ST 383)	K14 after (ST 383)	K15 before (ST 147)	K15 after (ST 147)	K16 after (ST 383)	K16 before (ST 383)
***aph*(*3′*)*-Ia***	Aminoglycoside resistance	**+**	**+**	**+**	**+**	**_**	**_**	**+**	**+**	**+**	**+**
***aac*(*6′*)*Ib-cr***	Fluoroquinolone And aminoglycoside resistance	**_**	**_**	**_**	**_**	**_**	**_**	**+**	**+**	**+**	**+**
***aph*(*3′*)*-VIa***	Aminoglycoside resistance	**_**	**_**	**_**	**_**	**_**	**_**	**_**	**_**	**+**	**+**
***aph*(*3′*)*-VIb***	Aminoglycoside resistance	**_**	**_**	**_**	**_**	**+**	**+**	**_**	**_**	**_**	**_**
** *aadA1* **	Aminoglycoside resistance	**_**	**_**	**_**	**_**	**_**	**_**	**_**	**_**	**+**	**+**
** *armA* **	Aminoglycoside resistance	**+**	**+**	**+**	**+**	**_**	**_**	**+**	**+**	**+**	**+**
** *strA* **	Aminoglycoside resistance	**+**	**_**	**+**	**+**	**+**	**+**	**+**	**+**	**+**	**+**
** *strB* **	Aminoglycoside resistance	**+**	**_**	**+**	**+**	**+**	**+**	**+**	**+**	**+**	**+**
** *fosA* **	Fosfomycin resistance	**+**	**+**	**+**	**+**	**+**	**+**	**+**	**+**	**+**	**+**
** *sul1* **	Sulphonamide resistance	**+**	**+**	**+**	**+**	**_**	**_**	**+**	**+**	**+**	**+**
** *sul2* **	Sulphonamide resistance	**+**	**+**	**+**	**+**	**_**	**_**	**+**	**+**	**+**	**+**
** *dfrA5* **	Trimethoprim resistance	**+**	**+**	**+**	**+**	**_**	**_**	**+**	**+**	**+**	**+**
***tet*(*A*)**	Tetracycline resistance	**_**	**_**	**_**	**_**	**+**	**+**	**_**	**_**	**+**	**+**
** *catA1* **	Phenicol resistance	**_**	**_**	**_**	**_**	**+**	**+**	**_**	**_**	**_**	**_**
** *oqxA* **	Quinolone resistance	**+**	**+**	**+**	**+**	**+**	**+**	**+**	**+**	**+**	**+**
** *oqxB* **	Quinolone	**+**	**+**	**+**	**+**	**+**	**+**	**+**	**+**	**+**	**+**
** *QnrS1* **	Quinolone resistance	**_**	**_**	**_**	**_**	**_**	**_**	**+**	**+**	**+**	**+**
** *QnrB1* **	Quinolone resistance	**_**	**_**	**_**	**_**	**_**	**_**	**_**	**_**	**+**	**+**
** *bla_CTX-M-14b_* **	Betalactam resistance	**+**	**_**	**+**	**__**	**+**	**+**	**+**	**+**	**+**	**+**
** *bla_CTX-M-15_* **	Betalactam resistance	**+**	**+**	**+**	**+**	**_**	**_**	**+**	**+**	**+**	**+**
** *bla_TEM-1B_* **	Betalactam resistance	**_**	**_**	**_**	**_**	**_**	**_**	**_**	**_**	**+**	**+**
** *bla_OXA-48_* **	Betalactam resistance	**+**	**__**	**+**	**_**	**+**	**+**	**+**	**+**	**+**	**+**
** *bla_SHV-1_* **	Betalactam resistance	**+**	**+**	**+**	**+**	**+**	**+**	**_**	**_**	**+**	**+**
** *bla_SHV-12_* **	Betalactam resistance	**_**	**_**	**_**	**_**	**_**	**_**	**+**	**+**	**_**	**_**
** *bla_NDM-5_* **	Betalactam resistance	**_**	**_**	**_**	**_**	**_**	**_**	**_**	**_**	**+**	**+**
** *bla_OXA-9_* **	Betalactam resistance	**_**	**_**	**_**	**_**	**_**	**_**	**_**	**_**	**+**	**+**
** *catA1* **	Phenicol resistance	**_**	**_**	**_**	**_**	**_**	**_**	**_**	**_**	**+**	**+**
** *Plasmid replicons* **		**K5** **before** **(ST 101)**	**K5** **after** **(ST 101)**	**K92** **before** **(ST 101)**	**K92** **after** **(ST 101)**	**K14** **before** **(ST 383)**	**K14** **after** **(ST 383)**	**K15** **before** **(ST 147)**	**K15** **after** **(ST 147)**	**K16** **after** **(ST 383)**	**K16** **before** **(ST 383)**
***IncFIB*(*Mar*)**		**+**	**+**	**+**	**+**	**_**	**_**	**+**	**+**	**+**	**+**
***IncL/M*(*pOXA- 48*)**		**+**	**__**	**+**	**_**	**+**	**+**	**+**	**+**	**+**	**+**
***IncFIB*(*pKPHS1*)**		**+**	**+**	**+**	**+**	**+**	**+**	**+**	**+**	**_**	**_**
** *IncHI1B* **		**+**	**+**	**+**	**+**	**_**	**_**	**+**	**+**	**+**	**+**
***IncFII*(*pKPX1*)**		**_**	**_**	**_**	**_**	**_**	**_**	**+**	**+**	**_**	**_**
***IncFIB*(*pQil*)**		**_**	**_**	**_**	**_**	**_**	**_**	**_**	**_**	**+**	**+**
***IncFII*(*K*)**		**_**	**_**	**_**	**_**	**_**	**_**	**_**	**_**	**+**	**+**

**Table 5 molecules-27-02541-t005:** Weight of fractions separated from liquid chromatography.

Fraction	Weight in mg
Fraction 1	4.31 mg
Fraction 2	17 mg
Fraction 3	20 mg
Fraction 4	16 mg
Fraction 5	31 mg
Fraction 6	33 mg
Fraction 7	62 mg

## Data Availability

Not applicable.

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
