# Peer review of "In Vitro Investigation of the Impact of Bacterial–Fungal Interaction on Carbapenem-Resistant Klebsiella pneumoniae"

_molecules, 2022, doi:10.3390/molecules27082541_

Round 1

Reviewer 1 Report

Main comments

   This manuscript describes about co-culture of fugal and antimicrobial resistant bacteria to induce the production of seconardy metabolites and it’s potential treatment for Carbapenem resistant Klebsiella pneumonia. Manuscript was poorely written and it need to be reviewed carefully. Overall, this manuscript is a primary identification of secondary metabolites and need to undergo many confirmation experiments for their claim.

Questions

  1. How did author confirm the loss of genes encoding virulence factor? Explain the techniques/experiments used
  2. Please discuss more theraputic details for the compound identified from GC-MS such as 2,4-Di-tert-butylphenol, 2,3-Butanediol, 9-octadecenamide and 11-octadecenoic acid in discussion section.
  3. Did author purified the indivual compound from the extract and identified the anti-bacterial efficiency?
  4. Please mention number of days the pneumonia got resistant with the secondary metabolites. How many passages does the resistant was stable?
  5. Show the GC-MS chromotograph picture for the metabolites identified
  6. What it mean by co-culture? Did author co-cultured in liquid media or Agar plate? How does the author identified the right growth temperature for both fungal and bacteria?
  7. Abstract and Conclusion section needs improvement. Please highlight the important findings of the manuscript.

Author Response

Reply to reviewer is uploaded

Reviewer 2 Report

The article of Moubasher Hani and co-workers discuss screened in co-culture with carbapenem resistant Klebsiella pneumoniae at different temperatures for inhibition of bacterial growth. Genotypic comparison of the K. pneumonia before and after co-culture was performed using next generation sequencing (NGS). Although the results are interesting and contribute to future alternatives for developing new antimicrobial susceptibility testing for K. pneumonia was done to compare antibiotic resistant patterns before and after co-culture. Overall, authors should complete missing information on experimental methods,  increase number of biological replicates in experimental design, major changes in the context/connection of results and discussion section along with improved presentation of results.
Major comments:

  • Title need to rewrite to be more attractive ex( In vitro investigation of the impact of bacterial-fungal interaction on Carbapenem-resistant Klebsiella pneumonia)
  • in authors name, please remove and between Elkholy Amani 
    and Sherif May
  • The manuscript without line numbers. this stranger for me. How add my comments?
  • Abstract need to rewrite.
  • Key words, just four words andare not intrest. What is mean by  co-culture!, where is other keywords such as: bacterial-fungal interaction; Carbapenem-resistant;  Klebsiella pneumonia
  • Improve the introduction section, please check it and improve the grammatical English.
  • Normally aim of the work at the end of introduction? where is aim of the study?
  • Ethical considerations at the end of Materials and methods. 
  • The study protocol was approved by the Research Ethics Committee of Faculty of Medicine, Cairo University, Egypt inaccordance with the Declaration of Helsinki (Ethical approval number: N-13-2020).
  • You could perform this research by using a reference strain?
  • 2.2. Isolation, screening of fungi with activity against K. pneumoniae
    2.3. Isolation of fungi from the soil.
    2.4. Isolation of fungi from hospital environment
    2.5. Identification of Fungal isolates
    Four section, each one contain one sentence? Put in one section.
  • You need to add phylogentic tree for your isolates after gentic analysis.
  • Figure 3 and 4 hand writing, please change computer writing.
  • Add chart of Liquid chromatography.
  • On overall, the sections 3 and 4 need to be improved, specially, the discussion of the results. Correlation of  co-culture with carbapenem resistant Klebsiella pneumoniae at different temperatures for inhibition of bacterial growth can be very helpful (supported by the findings of other authors)
  • The conclusion is too general, it should be connected and supported with the results.
  • please revise all name of bacteria in the text and change it to italic
  • References not in journal style

Some points should be revised, and the paper cannot be published in the current form in Journal as Molecules . Overall, authors should complete missing information on experimental methods and improve editing the article.

Author Response

Reply to reviewer is uploaded

Round 2

Reviewer 1 Report

Author revised the manuscript and provided the supplementary files and made changes in the abstract and conclusions. 

Revision is satisfactory. Thank you. 

Author Response

Thank you very much for all your kind recommendations which improved our paper so much. 

My regards

Hani Moubasher

Reviewer 2 Report

The change is very little and article not improved. It is still contain major revision. Authors not answer all comments:

  • The manuscript again without line numbers. this stranger for me. How add my comments?
  • Abstract need to rewrite again the cgange in it very little.
  • introduction not Improved, just aim added at the end of introduction>
  • Please check it and improve the grammatical English. The lines of article not justifies.
  • Ethical considerations without titleThis major revision not changed:
  • You need to add phylogentic tree for your isolates after gentic analysis.
  • Figure 3 and 4 hand writing, please change computer writing.
  • Add chart of Liquid chromatography.

Author Response

Dear Sir

Thank you very much for all your kind recommendations which improved our paper so much. We look forward to publishing our paper in your prestigious journal . We have done our best to take your recommendations into account and what we think needed to be changed. Thank you very much.

My regards

Hani Moubasher